# Uncertainty and Well-Being amongst Homeworkers in the COVID-19 Pandemic: A Longitudinal Study of University Staff

**DOI:** 10.3390/ijerph191610435

**Published:** 2022-08-22

**Authors:** Stephen Wood, George Michaelides, Kevin Daniels, Karen Niven

**Affiliations:** 1School of Business, University of Leicester, Brookfield, London Road, Leicester LE2 1RQ, UK; 2Norwich Business School, University of East Anglia, Norwich NR4 7TJ, UK; 3Sheffield University Management School, University of Sheffield, Sheffield S10 1FL, UK

**Keywords:** uncertainty, well-being, homeworking, COVID-19, job quality

## Abstract

The COVID-19 pandemic heightened uncertainties in people’s lives—and was itself a source of fresh uncertainty. We report a study of homeworkers on whether such uncertainties, and particularly those related to their work environment, are associated with lower levels of well-being and whether this association is exacerbated by prior poor well-being. We focus on five uncertainties surrounding the pandemic and employment—the virus, the job quality, workload, logistics of work lives, and support from the employer. Our empirical tests show that uncertainties around the virus, employer support, and their job quality have the strongest negative associations with well-being. These are based on data collected over three time periods in the first year of the pandemic from a sample of university staff (academics and non-academics) and well-being is measured on two continua, anxiety–contentment and depression–enthusiasm. The effects of uncertainties around workload and logistics are less pronounced, but more apparent among employees with better (not poorer) past well-being, at various times of the recession. The study adds to our understanding of the pandemic and highlights the need to link uncertainty to mental health more than it has in the past.

## 1. Introduction


*“I worked and completed many things [this week] but the very feeling of being confined at home and the uncertainty surrounding everything distracted me a lot and I felt down much of the time”. University employee, a respondent to a survey on home working in the pandemic.*


This quotation is from a respondent to a survey of homeworkers that we conducted in the initial phase of the COVID-19 when England was in lockdown. The “everything” in the quote could embrace a variety of domains. As Vieira, Potrich, Bressan, Klein, Pereira, and Pinto [1] (p. 7) say, the risk perceptions that underlie people’s uncertainties are a “function of individual reactions” in such domains; consequently, they are multidimensional and “there is no standard measure of risk perceptions that can be applied across hazards”. In this paper, we examine and develop a measure of the uncertainty facing homeworkers in the pandemic, covering uncertainties pertaining to the spread of the virus, forced homeworking and quarantining, availability of support, and aspects of their work. Using this measure, we assess the extent to which the various types of uncertainty contributed to homeworkers’ well-being during the first year of the pandemic, particularly contrasting the effects of uncertainties connected directly to COVID-19 to those related to their work and the support given by their employer.

Enforced and mass quarantining can be effective in reducing the spread of infectious disease, but it may result in psychological distress and impoverished psychological well-being, and the uncertainties surrounding pandemics may be significant contributor to this. Such effects may last well beyond the worst of a pandemic, estimated to be up to three years in one study [2]. In addition, given the rarity of such world-wide epidemics, people are less likely to have built-up resilience than they may be able to have when confronted more regularly with traumatic events such as physical abuse. With further outbreaks of COVID-19 continuing across the globe, as well as the advent of further infectious diseases, it is important to understand the role of uncertainty, and particularly the most significant domains of uncertainty, when assessing the impact of lockdowns and other public health interventions on mental health. In the case of homeworkers during the pandemic, a key question is whether it is the uncertainty surrounding the pandemic and other collective issues or the more personal proximal uncertainties concerning the logistics and nature of their work that have the strongest bearing on their well-being. It is important to know the ranking of these in order to know where information should be targeted to reduce uncertainty and who should be carrying out the communication and clarifications.

The study we report is focused on working adults who were able to work at home during the COVID-19 pandemic, thus helping to enact a key element of governments’ strategy to mitigate COVID-19. The uncertainties we will investigate are specific to workers: the virus, job quality, workload, logistics of work lives, and support from the employer. We first introduce these dimensions and then outline why we might expect uncertainty to have detrimental effects on well-being and reactions to uncertainty to be influenced by the effect of past uncertainty. We then report a study that first develops measures of the five dimensions of uncertainty and then assesses through factor analysis if data on these confirm they are discrete, before testing, using structural equation modeling, the relationship between the types of uncertainty and well-being. Finally, using relative weights analysis, we answer the open-ended question of which, if any uncertainty, may have the strongest association with well-being.

The study is longitudinal, using data from three time periods during the COVID-19 pandemic, two in 2020 and one in early 2021, collected from academic and professional service employees in two universities in England, with an initial sample of 584, and subsequent samples of 394, and 384. Uncertainty about the virus is global, but inconsistences, delayed responses, indecision, and mixed messages have been readily apparent in the UK context and indeed have become subject to intense criticism by the media, Parliamentary committees and within Parliament itself. As elsewhere, the uncertainty and its creation have become part of the story of COVID-19 in UK.

The research contributes to literature on uncertainty, by introducing a multi-factor measure of uncertainty for use in the work setting in pandemics that may be adapted to future such public health (or economic or political) crises. As well as providing evidence that uncertainty around the pandemic has consequences for people’s psychological well-being, the study therefore supplements existing measures of pandemic risk perception [1]. It further contributes by adding to the empirical examination of uncertainty’s link with well-being and particularly by introducing a dynamic element to understanding the psychological impacts of uncertainty, that takes account of how people’s prior well-being might shape their response to current uncertainties. It also adds to the literature on the COVID-19 pandemic in which people’s uncertainties have been all too apparent, but this has yet to be given much attention in the research, medical, or psychological domains (an exception is Charoensukmongkol and Phungsoonthorn [3]).

## 2. Assessing Uncertainty

### 2.1. Uncertainty

Uncertainty is a future-oriented cognition that is a state of mind rather than a feature of the objective world [4] (p. 830). We adopt Anderson, Carleton, Deifenach, and Han’s [5] (p. 2) definition of uncertainty as “the conscious awareness, or subjective *experience* of ignorance”, which highlights how the state differs from ignorance because it entails some level of awareness if it is to “influence [people’s] thoughts, feelings or actions”. Mishel [6] conceptualises the subjective experience of uncertainty as involving a sense of unpredictability and lack of clarity surrounding future events. For example, in the medical area, uncertainty arises from firstly the indeterminacy of the disease and secondly from the lack of clarity about the treatment(s). In an employment context, people might similarly experience uncertainty around unpredictable future events, such as not knowing whether their workload will change or how secure jobs will be in the future, alongside a lack of clarity about these and others matters in the communications and policies they receive from management. We concentrate on the uncertainty surrounding factors that may affect people’s ability to achieve work-related goals and balance work and home life, both of which are primary drivers of well-being [7,8,9,10].

We characterise the uncertainty surrounding homeworking during the pandemic in terms of five elements which are salient and relevant to employees. The first is the uncertainty concerning the virus itself and the lack of certainty about how it would progress, how safe public places might be, and what the future actions of the government in relation to the virus might be. This element of uncertainty pertains to the wider context in which work is embedded, in contrast to the other four aspects of uncertainty that we include which are more work-related. The second aspect of uncertainty that we consider concerns the nature and stability of people’s jobs in terms of what their work will entail and the tasks that ought to be prioritised (i.e., job uncertainty). The third element concerns uncertainty around workload, the level of demands people will face, the amount of time that will be required to complete tasks, and whether the demands they will face will feel excessive (i.e., workload uncertainty). The fourth aspect of uncertainty relates to logistics around people’s work lives, in particular regarding the use of homeworking to mitigate the pandemic and concerns about the extent to which work will be at home or on-site (logistics uncertainty). Finally, we include uncertainties about the level of support the employer will provide, for example the provision of information and communication and support for mental health (employer support uncertainty). We developed an instrument to measure each dimension with three items for each. These are presented in Table 1.

### 2.2. Uncertainty and Well-Being

Uncertainty’s effect on well-being has long been recognised but literature on this is diffuse. A theory that is applicable to understanding its link is the appraisal theory of psychological stress, which is a transactional theory in which well-being is framed a product of a transaction between individuals and their environment [11]. The core of the theory is that reacting to events involves an appraisal process, which explains why people’s affective responses to similar situations varies. Appraisal is an “evaluation of what one’s relationship to the environment implies for personal well-being” [12] (p. 234). Consequently, events are appraised in terms of whether they are harmful or beneficial for the person and how they relate to their goals and beliefs. Appraisal involves two stages as individuals first appraise events to determine if they are relevant to their well-being by considering whether an event has the potential to affect their goals, beliefs, values, or intentions, and, if so, whether it is potentially threatening. This is known as the primary appraisal. In the second stage, the secondary appraisal, individuals evaluate the extent to which they are able to cope with the event. How individuals appraise an event will determine the extent to which, and how, it affects their well-being. For example, appraisals of an event as being threatening and difficult to handle will stimulate feelings such as worry and fear, while appraisals of losses that are difficult to cope with will elicit sadness and misery [13] (p. 50). In such cases, a strain response is activated, which underlies the experience of poor well-being.

A lack of certainty in one’s work environment or more broadly one’s life could be considered threatening. As Mishel [6] (p. 259) explains, “When an event is uncertain, it is evaluated as a threat because the individual is not able to obtain a clear-cut conception of what is to occur”. It also entails a loss of clarity and predictability. Thus, a primary threat-appraisal would be anticipated among people who experience uncertainty. Secondary appraisals of low coping potential are also highly likely to be generated by uncertainties: unpredictability is difficult to cope with precisely because its defining feature is not knowing what is going to happen and when [14]. Therefore, individuals do not know which course of action, if any, is best to mitigate threat. As such, uncertainties are likely to trigger a strain response and ultimately poorer well-being. In particular, they may generate feelings of fear, threat, and worry which are associated with the poor well-being indicator of anxiety, or feelings of sadness, helplessness, and loss of control which are associated with depression [11,15].

We conceptualise anxiety and depression as ends of two continua of well-being on the basis of the Circumplex model which distinguishes between two axes of well-being: (a) pleasure, from pleasant to unpleasant, and (b) activation, from activation to deactivation [16]. Anxiety–contentment and depression–enthusiasm can be conceptualised as combinations of different levels of pleasure and activation. The anxiety–contentment dimension ranges from high activation and unpleasant affect (e.g., tension) to a low activation and pleasant affect (e.g., calmness). Depression–enthusiasm, on the other hand, ranges from a low activation unpleasant state (e.g., sadness) to a high activation and pleasant one (e.g., cheerful). Both anxiety and depression are forms of strain that can emerge from the appraisal process. Though they may co-exist, they are distinct and can even be negatively related, as some people may show less anxiety as they become more depressed [13] (p. 245).

Uncertainty has been widely associated with anxiety. In some cases, the association is implied almost by definition. For example, Grupe and Nitschke [17] (p. 497) state that anticipating or “pre-viewing” the future induces anxiety largely because the future is intrinsically uncertain.” Affective well-being is thus impaired as anxiety is generated from fear of the unknown, arising from an inability to construct a definitive conception of what is occurring or may occur. However, more subtly it also arises from doubts about how well one can cope with the ambiguities and uncertainties in the situation [18].

Uncertainty’s link to depression has been less discussed. The object of uncertainty may be about a potential irrevocable loss such as a death or something that people are unable to restore such as losing their job through being laid-off. It may be something over which people have some control but the fact that one is uncertain about its future state creates a sense of a loss of control or agency. An example in the pandemic is the uncertainty about the extent of mobility in the future and how one will cope with the consequent physical and mental entrapment.

Regardless of its type, we expect uncertainty has an immediate effect on well-being. This is consistent with appraisal theory according to which the effects of stressor appraisal on well-being are transmitted rapidly through cognitive processing, so as to appear to be concurrent [13]. We thus first test this initial-impact model for each of the three time periods in our study, through the hypothesis:

**Hypothesis** **1:**
*There will be a positive relationship between uncertainties surrounding the COVID-19 pandemic with (a) anxiety–contentment and (b) depression–enthusiasm at times 1, 2, and 3.*


In addition, uncertainty may affect the change in well-being from one period to another, such that as a person experiences uncertainty, their well-being shows a decline from its prior levels. We thus test:

**Hypothesis** **2:**
*There will be a positive relationship between uncertainties surrounding COVID-19 pandemic and changes in (a) anxiety–contentment and (b) depression–enthusiasm from time 1 to time 2 and from time 2 to time 3.*


Just as cognitive processes are involved in the generation of well-being, well-being affects the process of cognition. Past levels of well-being may affect the appraisal of stressors, and in turn, reactions to them. Therefore, emotional well-being may influence how perceptions of uncertainty are processed. Anxiety and depression can make individuals vulnerable to the negative effects of future uncertainty. Or conversely, being calm or cheerful has the potential to generate resilience to uncertainty. Illustrative of this, Chandler-Jeanville et al. [19] (p. 9), in a French study, showed how the nurses of front-line nurses caring for COVID-19 patients “experienced psychological vulnerability, as they felt strained physically and emotionally, by their relentless exposure to the virus consequences”. This strengthened their death anxiety and weakened their mental health.

We expect that anxiety will direct people’s attention processes towards unhelpful information and thus that those who have experienced high levels of past anxiety will make different primary and secondary appraisals of the same level of uncertainty from those who have been more content [20,21]. Such appraisals will produce further anxiety. We expect a similar effect for depression as it impedes memory searching, with the effect that people are more likely to dwell on negative information and ruminate on negative experiences [20,21]. As a consequence, those who have experienced high levels of depression in the past are more likely to make negative appraisals when facing uncertainties than those who were more enthusiastic in the past. We thus expect an interaction effect between present uncertainty and past well-being on present well-being, that is the relationship between uncertainty on well-being at time 2 and 3 is strongest for those with the poorest well-being at time 1 and 2, respectively. We thus test:

**Hypothesis** **3:**
*Past anxiety–contentment/depression–enthusiasm (at time 1 and time 2) will moderate the effect of uncertainty on present anxiety–contentment/depression–enthusiasm (at time 2 and time 3) so that uncertainty will have a stronger (negative) effect on well-being for those with low prior well-being.*


It is unlikely that each dimension of uncertainty has the same strength of relationship with well-being. We leave the question of which dimensions may be the most influential open. For on the one hand, factors proximal to everyday working practices, which may be more disruptive of planning and predicting day-to-day work activities, may have the most effect. However, on the other hand, uncertainty around the virus and the government’s attempts at containment may be more significant as they out of an individual’s control and reflect potentially more serious personal and societal outcomes.

We thus address the following research question:


*Research Question 1: What is the ranking of the unique contribution of each type of uncertainty to well-being.*


## 3. Methodology

### 3.1. The Study

We tested our hypotheses using data collected from employees in two universities in England (one in the Midlands and one in the South), academics and non-academics, whilst working at home, at three phases during the first year of the pandemic, July and October 2020 and February 2021. The UK was in a full lockdown during the first and third waves of the survey in July 2020 and February 2021, with some easing of restrictions at the time of the second wave of the survey in October 2020. However, even during the second survey wave, national guidelines on appropriate behaviour existed, for example on social distancing, and additional local restrictions were in place in some places, including the location of one of the two universities. Working at home wherever possible remained the dictat from government in all periods, and both universities were closed for face-to-face teaching and all but essential activities throughout the study. Since the homeworking is enforced and total, our results are not compounded by the degree of time spent homeworking, which is relevant because previous research suggests low levels of homeworking may increase well-being while high levels diminish it [22].

### 3.2. The Sample

For the first data collection wave, there were 584 surveys where all uncertainty items were answered. For the second wave, this was 394, and 384 for the third wave. Each of these samples were used to validate the factor structure of the uncertainty measure. A total of 254 participants fully completed all three waves of the survey and their data were used for testing the hypotheses. A larger proportion of the three-wave sample was from the Midlands university (58.27%), female (77.17%), white (95.67%), non-academics (68.11%) and full-time employees (74.41%). The mean age of our sample was 45 years (SD = 11.06), with an age range from 22 to 73 years. Fifty-one per cent of academics in the sample reported working at home at least one day a week prior to the pandemic; in contrast, only eight per cent of non-academics had any history of working at home.

## 4. Measures

### 4.1. Uncertainty

The wording of the items for the five uncertainty measures aimed to get as close as possible to the evaluations of the participants. They were designed to capture uncertainty feelings or assessment of potential issues, not “cold cognitive estimates of probabilities” [17] (p. 492). We designed the items so that people were responding to situations they were experiencing in order to minimise the risk of their contriving responses to potentially hypothetical situations. We developed 15 items, 3 items pertaining to each source of uncertainty (Table 1) and asked “Thinking about the coming months, how certain or uncertain do you feel about the following [situation]”. We used an even-numbered eight-point scale without a mid-point, to avoid prevarication and reduce any tendencies to go for the neutral option or to the two extremes, thus: 1 = very uncertain, 2 = fairly uncertain, 3 = moderately uncertain, 4 = slightly uncertain, 5 = slightly certain, 6 = moderately certain, 7 = fairly certain, 8 = very certain. To create the measure of uncertainty, responses were reverse coded.

### 4.2. Well-Being

This is measured using a shortened eight-item version of Warr’s [23] scales of two continua from anxiety–contentment and depression–enthusiasm. Respondents were asked to rate the extent to which they experience different affective states in the last seven days on a five-point scale. The states for anxiety–contentment are “anxious”, “worried”, “at ease”, “relaxed”, and for depression–enthusiasm they are “depressed”, “gloomy”, “happy”, and “cheerful”. The five-point scale was: “never”, “occasionally”, “some of the time”, “most of the time”, and “all of the time”. Responses to positive well-being items were reverse coded, so that high scores represented poor well-being.

### 4.3. Control Variables

In our initial exploration of the data, we considered participants’ age, gender, tenure, position (academic/non-academic), and organisation (Southern /Midlands university) as potential control variables. Gender and tenure were unrelated to either measure of well-being for any of June, October, or February and were excluded from subsequent analyses.

## 5. Analysis Approach

We first assessed the factor structure of our uncertainty measures, using confirmatory factor analysis (CFA) to assess if the items used conformed with our expected factor structure of five uncertainty dimensions. For each of the three data collection periods, we evaluated the factor structure separately, allowing us to test for configural invariance. We then used a multigroup CFA to evaluate metric invariance by comparing a model with all parameters being estimated freely for the three surveys to one in which the factor loadings were held to be the same for the three surveys. We also evaluated scalar invariance; however, rather than exhibiting invariance, we expect that the intercepts will in fact change, thus reflecting that the instrument captures differences in the levels of uncertainty.

We then used the resulting factors, calculated as the item means, in structural equation models to evaluate our three hypotheses. We ensured that relationships from different data collections were treated independently to each other by constraining the covariance between them to be zero. The covariances between anxiety–contentment and depression–enthusiasm in each data collection were treated as free model parameters. For the first hypothesis, we controlled for age, position, and university, and for the second, we also included well-being at the previous data collection period (in order to capture change in well-being), so for October 2020 we controlled for July 2020 well-being and for February 2021 we controlled for October 2020 well-being. To evaluate the third hypothesis, we developed the model from the second hypothesis that contained past well-being by adding the interaction effects of past well-being and each of the uncertainty variables.

To evaluate research question 1, we use relative importance or weights analysis [24,25]. This is a method that assesses the amount of variance in the criterion variable that can be attributed to each predictor, in contrast to using the *R*^2^ which reflects the amount of variance that the predictors jointly explain. It can be used to supplement regression analysis where predictors share nontrivial amounts of variance. To evaluate the relative importance of the interaction effects, we used the residualised interactions recommended by LeBreton, Tonidandel, and Krasikova [26].

Analyses were performed with R 4.1.0 (R Core Team [27]) and the lavaan package [28]. The relative weight analysis was performed with the RWA package [29].

## 6. Results

### 6.1. The Uncertainty Scales and Descriptive Data

For each of the three data collection periods, we compared the five-factor CFA model to reflect the five types of uncertainty to a single-factor model that includes all the items. The first two items for logistics uncertainty had a high residual correlation that was not shared by the third item and they were therefore explicitly specified to correlate in all the models tested. As shown in Table 2, the results for each of the three time periods indicate that the five-factor model was a good fit and was significantly better than the single-factor model. The five-factor configuration is consistent across the three surveys thus showing configural invariance. Using multi-group CFA, the model was not significantly different to a model where the factor loadings were constrained to be the same in July, October, and February, thus showing metric invariance. Constraining the model intercepts did however produce significantly different results, thus indicating that there is no evidence for scalar invariance and that the five factors captured change in uncertainty over time.

Descriptive statistics and correlations for all variables used in the analysis are presented in Table 3.

Since some of the associations examined are between constructs assessed at the same time, there is a potential common method bias. To evaluate if this was indeed a problem in our case, we used the Harman single factor approach (see Podsakoff et al. [30]), which we tested separately for all the items in July 2020, October 2020, and February 2021. In each case, the results suggested that total variance explained by the combined factor of all the items was 32% for July 2020, 33% for October 2020 and 33% for February 2021. Since in each case the result was less than 50%, we can conclude that common method bias was not prevalent in any of the three data collection points. A test of all the items combined showed a similar result with the single factor explaining only 27% of the variance in the items.

### 6.2. Testing Hypothesis 1

For the first hypothesis, we tested if the five measures of uncertainty influenced anxiety–contentment and depression–enthusiasm in each of the three waves of the study. The results are shown in Table 4. There are significant effects for virus, job, and employer support uncertainty and all significant results are in the expected direction so that uncertainty is positively associated with poor well-being. Virus uncertainty was significantly related to anxiety–contentment in July 2020 and February 2021. Job uncertainty was related to anxiety–contentment in all three data collections and depression–enthusiasm in July and October 2020. Employer support uncertainty was associated with anxiety–contentment for all three data collection points and depression–enthusiasm in October 2020 and February 2021.

### 6.3. Testing Hypothesis 2

The second hypothesis was evaluated by controlling for well-being in the previous time period and thus we are able to evaluate if uncertainty is influencing changes in well-being (Table 5). The results for October 2020 and February 2021 show that only job and employer support uncertainties were related to changes in well-being. Job uncertainty predicted an increase in anxiety from the previous data collection in both October 2020 and February 2021 and an increase in depression in February 2021. This contrasts with the result from the test of the first hypothesis, when past well-being was not used as a control, as job uncertainty related to depression–enthusiasm in October 2020 but not in February 2021. Employer support uncertainty predicted an increase in anxiety from the previous data point in October 2020 and an increase in depression in both October 2020 and February 2021.

### 6.4. Testing Hypothesis 3

For the third hypothesis, we tested if poor well-being from the previous period exacerbated the associations between uncertainties and poor well-being. The results are shown in Table 6. Similar to the second hypothesis, we can only test this hypothesis for October 2020 and February 2021.

In terms of the interaction effects, there are significant results for workload uncertainty for October 2020 and logistics uncertainty in February 2021 for both anxiety–contentment (Figure 1) and depression–enthusiasm (Figure 2). However, the results were in the opposite direction to those hypothesised. That is, rather than finding a build-up of harm, so that those with poor well-being in the prior data collection were more negatively affected by current uncertainties, we found that uncertainty had a stronger negative effect for those who had better prior well-being.

### 6.5. Testing Research Question 1

The relative weights analysis confirmed that all five uncertainty variables contribute to both anxiety–contentment and depression–enthusiasm (Table 7). In particular, the form of uncertainty that seems to have the most substantial impact across all data waves is employer support uncertainty, with job uncertainty following closely behind. There was a steep drop-off in the relative impact of virus uncertainty after the first data wave.

## 7. Discussion

### 7.1. Overview

Uncertainty has characterised many workers’ experiences during the COVID-19 pandemic. Our study has highlighted how five specific aspects of uncertainty—pertaining to the virus, job quality, workload, logistics, and employer support—have shaped workers’ mental health during the early phases of the pandemic. Across the testing of our hypotheses, we found that all the aspects of uncertainty were associated with well-being at some stage in the period captured in the study, either having an immediate detrimental impact on well-being, or contributing towards a decline in well-being over time. The relevance of each type of uncertainty for well-being gives some confidence in the external validity of our novel measure, supplementing the internal reliability provided by the strong Cronbach alphas.

Our results showed that three forms of uncertainty were particularly potent for undermining employees’ immediate well-being through the pandemic. Virus, job, and employer support uncertainty were all associated with anxiety–contentment in all three periods, with only one exception, and with depression–enthusiasm in some time periods. Both job uncertainty and employer support uncertainty also predicted a worsening of well-being over time across the study period, indicating that these forms of uncertainty might be particularly salient in explaining declines in mental health among workers through the pandemic.

In contrast, workload and logistics uncertainty have a limited impact on well-being, and only for those with high past well-being, which is contrary to what we predicted on the basis of an assumed cumulative effect of poor well-being. These uncertainties have a detrimental effect on those with high past well-being. The timing of the significant interactions may be important to help interpret these findings. For workload uncertainty, this has a negative association with anxiety–contentment and depression–enthusiasm in October 2020 amongst those with good prior well-being. October coincides with the beginning of term in the UK university context and in 2021, staff were faced with fresh demands and the challenges of on-line and blended teaching, plentiful team meetings, and having work “dumped” on them at the last minute, as one respondent said in the comment box at the end of the October questionnaire. Our findings suggest that uncertainties surrounding these workload issues had a greater effect on staff members whose well-being was high in July 2020. The negative effect of logistics uncertainty on well-being in February 2021, which did not occur in October 2020, may reflect the fact that the country was again in lockdown whereas this was not the case in October. The uncertainty about when employees might return to on-site working and more normal caring arrangements would be especially salient when there were feelings of there being “little light at the end of the tunnel” and that home confinement was “never ending”.

The insignificance of virus uncertainty in October may be because England was not in a total lockdown as it was at the other times. Yet, while average levels of virus uncertainty reduced from their July level, this cannot explain the lack of a link. The same levels of uncertainty may however have less of an effect on well-being in October than it did in July as people interpreted the ending of the lockdown as a sign that the virus was on the way out and further restrictions would be lifted and uncertainty about the pandemic’s future was psychologically manageable.

The differential results between anxiety–contentment and depression–enthusiasm reinforce arguments that they are distinct [13] (p. 245), though the past levels of each both played a role in the moderation of the effect of workload and logistics uncertainty. That the main effects of the other types of uncertainty were stronger for anxiety–contentment may reflect the way they generate more fear or threats to one’s resources, rather than a sadness or guilt about not being able to accomplish demands.

### 7.2. Strengths, Limitations and Future Research

Strengths of the study include its longitudinal design which enabled us to test the effects of prior well-being on subsequent well-being, and its coverage of significant areas of uncertainty in the pandemic. Uncertainties are domain-specific. Our methodology is illustrative of how one might further the study of uncertainty and extend coverage to other areas. All but the virus uncertainty scale could be used or adapted in future studies in the work and work–home interface areas. The vulnerability of university homeworkers to the pandemic and its economic effects was perhaps as low as any group in the economy, and thus since we find uncertainty plays a role in their well-being, we might expect it to have played a similar role in other groups. The extent to which the results involving uncertainties other than virus uncertainty are affected by the pandemic or homeworking context is unclear. This is particularly so in the case of the two results that support Hypothesis 3: the interaction effects involving workload and logistics uncertainty. However, the empirical study offers support for our underlying theoretical foundations and suggests it is a promising line of enquiry: the impact of uncertainty on well-being may be a generalisable phenomenon.

The use of self-report measures for both uncertainty and well-being was necessary as they involve internal or intrapsychic processes; self-reporting provides the most direct and reliable assessment method. However, this has the potential to create common method variance and enhance the size of the associations we estimated. Nonetheless, our analysis involved hypotheses with data collected at different time points making the possibility for common method variance less likely. The presence of interaction effects is particularly salient in this respect as these are unlikely to be artifacts of common method variance and are more difficult to detect if there is common method variance [31]. More importantly, using the Harman single-factor test we found no evidence that common method variance was an issue for concern for the items measured at the same time.

While we present an initial validation of our new measure of work-related uncertainty, future research could validate it against a broader range of criteria, for example, to demonstrate convergent validity with personality characteristics associated with the experience of uncertainty, such as neuroticism.

An addition to future research could be to include personality measures such as neuroticism or locus of control as moderators of relationships. Typically, an uncertain situation or event is, in Mischel’s [32] terms, a weak situation and hence the scope for individual differences in reactions to them is large. The variety in uncertainty ratings shows that situations are not uniformly encoded and do not induce common responses. Such “weak” situations are where personality is likely to play its greatest role in explaining well-being and cognitions. We thus anticipate it will moderate relationships between uncertainty and well-being. We might also conjecture that if it does not, it reflects a state where the uncertainty situation is strong, which on the surface may be a contradictory term. Nonetheless, where national or organisational cultures are strong, this may be feasible. For example, in centralised states such as China, a common perspective on the pandemic may have been fermented so individual judgements of its future course might be uniform.

### 7.3. Practical Implications

Our findings have implications for employers, policy makers, and individuals in how to address issues related to employee well-being during future pandemics and more widely. For employers, the COVID-19 pandemic required many organisations to adopt radically different working practices and simultaneously address significant changes in employees’ concerns about their well-being (see e.g., Nayani et al. [33]). Our findings indicate that acting to reduce uncertainty over working practices and employer support are two areas in which organisations could target to protect employee well-being. This could include extensive use of two-way communication with employees, so that employers are aware of the specific concerns of employee groups and so can act collaboratively to address these concerns [33]. Doing so has the added advantage of conveying genuine care for employers. Uncertainty over employer support during crises may also be reduced by consistent and evolving programmes of occupational health and well-being initiatives during times of stability. Continuous improvement in workplace occupational health/well-being programmes has potential dual benefits: (i) it may develop organisational capabilities that facilitate adaptation to major outbreaks of infectious diseases when they occur, and (ii) it should inculcate a shared perception that the organisation will support employee health and well-being whatever the circumstances [34]. 

Following the study of Richer et al. [35] on how to convey bad or difficult news, another approach that may be helpful for employers to reduce employees’ uncertainty could focus on communicating as factually-correct information as possible in a fair way. Their study highlighted that giving managers targeted training can pay dividends. The implication is that how one communicates and by what means is important, but the nature of what one communicates is crucial; vagueness, obfuscation, and incomplete information is to be avoided. Just as employees’ strain reactions are based on both gauging events and how they will cope with them, so employers must consider describing both events or change and what their policies and practices will be. This is akin to a medic relaying the diagnosis and the treatment. 

For policy makers, our findings suggest that local and central governments provide guidance on how best to implement infection control measures in workplaces and details of possible time courses and consequences in order to help employers manage transitions to new and temporary working practices. The findings also suggest policy makers should broadcast statistics and projections about the pandemic, as countries have done, and offer specific and clear prescriptions on appropriate individual behaviour to reduce uncertainty perceptions about the virus itself, and this protect well-being in the wider population.

At the individual level, our findings suggest that employees may actively seek information and changes in their environment as the job crafting approach implies. The evidence of its effects on well-being are accumulating [36,37] and interventions can help individual’s job crafting [38].

## 8. Conclusions

The uncertainty around the COVID-19 pandemic and the reported negative effects on mental have highlighted the neglect of uncertainty in the study of well-being and psychological strain. We have shown that there are strong grounds for linking uncertainty to anxiety–contentment and depression–enthusiasm. In particular, the level of uncertainty in a domain will affect concurrent well-being and in certain cases, the effect will be moderated by past well-being. Conceptions of uncertainty are domain-specific and have to be identified by the form of the unknown, thus developing generic measures of uncertainty are unlikely to be fruitful. Our measures are illustrative of how domain-specific scales can be developed and we hope will inspire future work. Their development and application in the COVID-19 pandemic show that uncertainty played a role in the well-being of workers in the pandemic.

## Figures and Tables

**Figure 1 ijerph-19-10435-f001:**
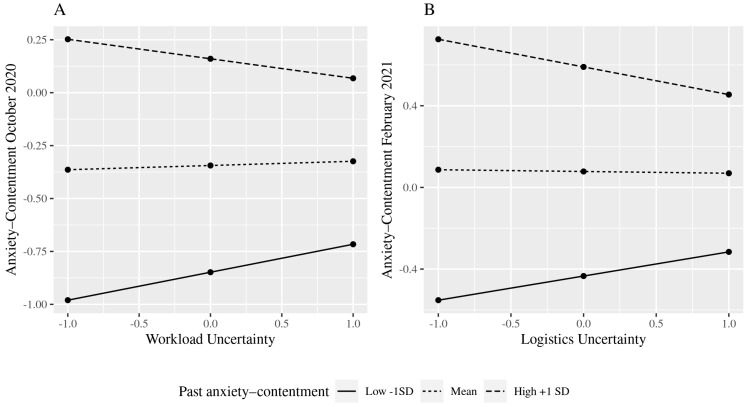
(**A**,**B**) Interaction effects between past anxiety–contentment and uncertainty.

**Figure 2 ijerph-19-10435-f002:**
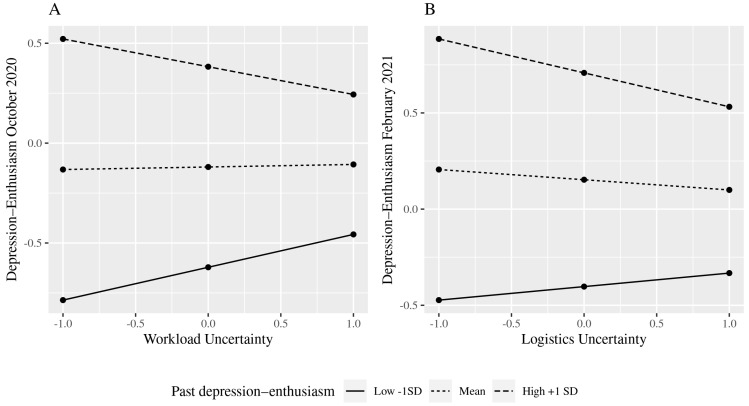
(**A**,**B**) Interaction effects between past depression–enthusiasm and uncertainty.

**Table 1 ijerph-19-10435-t001:** The new uncertainty scale with unstandardised factor loadings for the different study phases.

Thinking about the Coming Months, How Certain or Uncertain Do You Feel about the Following?	July 2020	October 2020	February 2021
*Job Uncertainty*			
What my job is going to involve	1.00	1.00	1.00
What tasks I will be expected to prioritise	1.02	0.99	1.05
How secure my job will be	0.67	0.75	0.71
*Workload Uncertainty*			
How hard I will be required to work to get my job done	1.00	1.00	1.00
The amount of hours I will need to work to complete my work tasks	1.00	0.94	1.09
The extent to which I will be asked to do an excessive amount of work	0.80	0.83	0.92
*Logistics Uncertainty*			
What proportion of my work time I will work at home	1.00	1.00	1.00
What proportion of my work time I will work on campus	0.93	0.98	1.02
Whether I will be able to balance work and non–work responsibilities	1.08	1.09	1.03
*Support Uncertainty*			
Whether I will be provided access to the equipment and services I need in order to do my job (e.g., IT hardware and software, IT support)	1.00	1.00	1.00
Whether the University will support my well-being	1.06	1.44	1.42
Whether the University will protect my personal safety while at work	0.84	1.19	1.16
*Virus Uncertainty*			
How the pandemic is likely to progress	1.00	1.00	1.00
How to judge how safe venues such as shops, pub and restaurants are	1.28	1.09	1.18
Whether the government will swiftly detect fresh outbreaks of COVID-19 cases	1.05	0.93	1.09

**Table 2 ijerph-19-10435-t002:** Measurement invariance for the five-factor uncertainty model.

	X^2^	df	ΔX^2^	CFI	TLI	Gamma Hat	RMSEA	SRMR
*July 2020*								
Five-factor model	324.53	79		0.95	0.93	0.77	0.07	0.05
Single-factor model	1425.29	89	1100.76 ***	0.72	0.66	0.95	0.16	0.11
*October 2020*								
Five-factor model	263.96	79		0.94	0.92	0.76	0.08	0.06
Single-factor model	1009.97	89	746.01 ***	0.70	0.64	0.94	0.16	0.10
*February 2021*								
Five-factor model	214.59	79		0.96	0.94	0.78	0.07	0.05
Single-factor model	928.20	89	713.61 ***	0.74	0.69	0.96	0.16	0.11
*Invariance models*								
Configural	803.07	237		0.95	0.93	0.94	0.07	0.05
Metric	832.00	257	28.93	0.95	0.94	0.95	0.07	0.06
Scalar	894.36	277	62.36 ***	0.94	0.93	0.94	0.07	0.06

*** *p* < 0.001.

**Table 3 ijerph-19-10435-t003:** Descriptive statistics and inter-correlations between the main study variables among the three-wave sample (*N* = 254).

	M	SD	α	1	2	3	4	5	6	7	8	9	10	11	12	13	14	15	16	17	18	19	20
1. Anxiety–Contentment—July	3.12	0.81	0.88																				
2. Depression–Enthusiasm—July	3.58	0.76	0.88	0.73																			
3. Virus Uncertainty —July	5.98	1.43	0.74	−0.29	−0.22																		
4. Job Uncertainty—July	4.41	1.77	0.84	−0.30	−0.35	0.32																	
5. Workload Uncertainty—July	3.59	1.81	0.82	−0.23	−0.30	0.27	0.70																
6. Logistics Uncertainty—July	4.58	1.88	0.73	0.17	0.23	−0.27	−0.48	−0.54															
7. Support Uncertainty—July	3.41	1.58	0.74	−0.27	−0.29	0.36	0.41	0.39	−0.46														
8. Anxiety–Contentment—Oct	3.03	0.83	0.88	0.66	0.59	−0.26	−0.36	−0.26	0.21	−0.35													
9. Depression–Enthusiasm—Oct	3.51	0.77	0.84	0.53	0.67	−0.25	−0.29	−0.25	0.22	−0.32	0.76												
10. Virus Uncertainty—Oct	5.81	1.47	0.72	−0.22	−0.15	0.59	0.29	0.29	−0.25	0.29	−0.29	−0.27											
11. Job Uncertainty—Oct	4.33	1.85	0.84	−0.29	−0.33	0.32	0.65	0.45	−0.34	0.33	−0.41	−0.38	0.41										
12. Workload Uncertainty—Oct	3.32	1.73	0.86	−0.17	−0.28	0.25	0.53	0.57	−0.30	0.34	−0.28	−0.30	0.34	0.71									
13. Logistics Uncertainty—Oct	5.23	1.87	0.72	0.15	0.17	−0.29	−0.37	−0.35	0.52	−0.39	0.27	0.29	−0.37	−0.49	−0.42								
14. Support Uncertainty—Oct	3.43	1.69	0.81	−0.29	−0.32	0.32	0.41	0.37	−0.35	0.68	−0.43	−0.43	0.41	0.53	0.48	−0.53							
15. Anxiety–Contentment—Feb	2.98	0.83	0.89	0.54	0.49	−0.29	−0.28	−0.24	0.25	−0.29	0.65	0.56	−0.24	−0.35	−0.18	0.22	−0.31						
16. Depression–Enthusiasm—Feb	3.43	0.81	0.88	0.46	0.60	−0.25	−0.30	−0.24	0.15	−0.18	0.56	0.69	−0.25	−0.37	−0.23	0.18	−0.31	0.72					
17. Virus Uncertainty—Feb	5.66	1.47	0.71	−0.33	−0.27	0.60	0.30	0.28	−0.25	0.29	−0.36	−0.32	0.61	0.36	0.28	−0.32	0.32	−0.37	−0.35				
18. Job Uncertainty—Feb	4.28	1.87	0.85	−0.25	−0.23	0.25	0.51	0.32	−0.20	0.21	−0.30	−0.21	0.22	0.58	0.37	−0.28	0.30	−0.38	−0.34	0.38			
19. Workload Uncertainty—Feb	3.37	1.78	0.88	−0.14	−0.14	0.21	0.41	0.42	−0.30	0.23	−0.18	−0.18	0.29	0.50	0.50	−0.36	0.35	−0.27	−0.30	0.34	0.71		
20. Logistics Uncertainty—Feb	4.74	1.86	0.73	0.06	0.11	−0.31	−0.34	−0.27	0.52	−0.29	0.18	0.19	−0.30	−0.30	−0.22	0.50	−0.29	0.22	0.20	−0.39	−0.37	−0.44	
21. Support Uncertainty—Feb	3.48	1.55	0.75	−0.25	−0.27	0.33	0.36	0.26	−0.31	0.55	−0.36	−0.34	0.32	0.40	0.27	−0.39	0.67	−0.35	−0.31	0.42	0.50	0.46	−0.50

Values above 0.12 or below −0.12 are significant at *p* = 0.05.

**Table 4 ijerph-19-10435-t004:** Effects of uncertainty on well-being (Hypothesis 1).

	July 2020			October 2020			February 2021		
	B		SE	B		SE	B		SE
*Anxiety–contentment*									
Age	0.02	**	0.01	0.02	***	0.00	0.01	*	0.00
Southern (0)/Midland university (1)	−0.06		0.12	−0.03		0.11	−0.06		0.11
Academic (0)/Nonacademic (1)	−0.03		0.12	−0.04		0.12	−0.07		0.12
Virus Uncertainty	−0.18	**	0.06	−0.05		0.06	−0.21	**	0.06
Job Uncertainty	−0.24	**	0.09	−0.32	***	0.08	−0.23	**	0.08
Workload Uncertainty	0.03		0.09	0.10		0.08	0.04		0.08
Logistics Uncertainty	−0.03		0.07	−0.02		0.07	−0.02		0.07
Support Uncertainty	−0.15	*	0.07	−0.31	***	0.08	−0.18	*	0.08
*Depression–Enthusiasm*									
Age	0.01		0.00	0.01	**	0.00	0.01	*	0.00
Southern (0)/Midland university (1)	−0.27	*	0.11	−0.10		0.11	−0.08		0.11
Academic (0)/Nonacademic (1)	−0.02		0.12	−0.17		0.12	−0.23		0.12
Virus Uncertainty	−0.07		0.06	−0.03		0.06	−0.19	**	0.06
Job Uncertainty	−0.23	**	0.08	−0.19	*	0.08	−0.14		0.08
Workload Uncertainty	−0.05		0.08	0.01		0.08	−0.07		0.08
Logistics Uncertainty	0.04		0.07	0.03		0.07	−0.03		0.07
Support Uncertainty	−0.13		0.07	−0.32	***	0.08	−0.18	*	0.08

* *p* < 0.05, ** *p* < 0.01, *** *p* < 0.001.

**Table 5 ijerph-19-10435-t005:** Effects of uncertainty on changes in well-being in July, October, and February.

	July			October			February		
	B		SE	B		SE	B		SE
*Anxiety–contentment*									
Age	0.02	**	0.01	0.01	*	0.00	0.00		0.00
Southern (0)/Midlands university (1)	−0.06		0.12	−0.02		0.09	−0.04		0.09
Academic (0)/Nonacademic (1)	−0.03		0.12	−0.02		0.10	−0.08		0.10
Virus Uncertainty	−0.18	**	0.06	−0.02		0.05	−0.09		0.05
Job Content Uncertainty	−0.24	**	0.09	−0.18	**	0.07	−0.13	*	0.07
Job Demands Uncertainty	0.03		0.09	0.05		0.06	−0.02		0.07
Logistics Uncertainty	−0.03		0.07	0.01		0.05	0.01		0.05
Support Uncertainty	−0.15	*	0.07	−0.20	**	0.06	−0.04		0.07
Past Anxiety–Contentment				0.51	***	0.04	0.51	***	0.05
*Depression–Enthusiasm*									
Age	0.01		0.00	0.01	*	0.00	0.01		0.00
Southern (0)/Midlands university (1)	−0.27	*	0.11	−0.02		0.09	−0.01		0.08
Academic (0)/Nonacademic (1)	−0.02		0.12	−0.16		0.10	−0.16		0.09
Virus Uncertainty	−0.07		0.06	−0.04		0.05	−0.08		0.05
Job Content Uncertainty	−0.23	**	0.08	−0.09		0.07	−0.13	*	0.06
Job Demands Uncertainty	−0.05		0.08	0.05		0.06	−0.08		0.06
Logistics Uncertainty	0.04		0.07	0.06		0.05	−0.03		0.05
Support Uncertainty	−0.13		0.07	−0.21	***	0.06	−0.02		0.06
Past Depression–Enthusiasm				0.53	***	0.04	0.56	***	0.04

* *p* < 0.05, ** *p* < 0.01, *** *p* < 0.001.

**Table 6 ijerph-19-10435-t006:** Effects of uncertainty on changes in well-being (Hypothesis 2) and interactions between uncertainty and past well-being (Hypothesis 3).

	July			October			February		
	B		SE	B		SE	B		SE
*Anxiety–contentment*									
Age	0.02	**	0.01	0.01	*	0.00	0.00		0.00
Southern (0)/Midlands university (1)	−0.07		0.12	−0.03		0.09	−0.06		0.09
Academic (0)/Nonacademic (1)	−0.05		0.12	0.00		0.10	−0.06		0.10
Virus Uncertainty	−0.17	**	0.06	−0.02		0.05	−0.12	*	0.05
Job Uncertainty	−0.23	**	0.09	−0.16	*	0.07	−0.11		0.07
Workload Uncertainty	0.02		0.09	0.02		0.07	−0.07		0.07
Logistics Uncertainty	−0.04		0.07	0.03		0.05	−0.01		0.06
Support Uncertainty	−0.15	*	0.07	−0.19	**	0.06	−0.02		0.07
Past Anxiety–Contentment				0.50	***	0.04	0.51	***	0.04
Past Anxiety–Contentment × Virus Uncertainty				0.03		0.04	0.05		0.05
Past Anxiety–Contentment × Job Uncertainty				0.01		0.06	0.08		0.06
Past Anxiety–Contentment × Workload Uncertainty				−0.11	*	0.06	−0.11		0.06
Past Anxiety–Contentment × Logistics Uncertainty				−0.06		0.05	−0.13	*	0.05
Past Anxiety–Contentment × Support Uncertainty				0.03		0.06	−0.03		0.06
*Depression–Enthusiasm*									
Age	0.01		0.00	0.01	*	0.00	0.01		0.00
Southern (0)/Midlands university (1)	−0.28	*	0.11	−0.04		0.09	−0.02		0.08
Academic (0)/Nonacademic (1)	−0.03		0.12	−0.14		0.10	−0.15		0.09
Virus Uncertainty	−0.07		0.06	−0.04		0.05	−0.09		0.05
Job Uncertainty	−0.22	**	0.08	−0.08		0.07	−0.09		0.06
Workload Uncertainty	−0.06		0.08	0.01		0.06	−0.13		0.07
Logistics Uncertainty	0.03		0.07	0.08		0.05	−0.05		0.05
Support Uncertainty	−0.12		0.07	−0.17	**	0.06	0.00		0.06
Past Depression–Enthusiasm				0.50	***	0.04	0.56	***	0.04
Past Depression–Enthusiasm × Virus Uncertainty				−0.04		0.05	0.03		0.05
Past Depression–Enthusiasm × Job Uncertainty				0.08		0.06	−0.02		0.06
Past Depression–Enthusiasm × Workload Uncertainty				−0.15	**	0.06	−0.10		0.05
Past Depression–Enthusiasm × Logistics Uncertainty				−0.08		0.05	−0.12	*	0.05
Past Depression–Enthusiasm × Support Uncertainty				0.08		0.05	0.04		0.05

* *p* < 0.05, ** *p* < 0.01, *** *p* < 0.001.

**Table 7 ijerph-19-10435-t007:** Relative Weight Analysis of uncertainty on well-being.

	Anxiety July	Depression July	Anxiety October	Depression October	Anxiety February	Depression February
Age	4.73	2.08	1.12	1.49	3.99	2.57
Southern (0)/Midlands university (1)	1.65	10.89	0.07	0.48	0.28	0.39
Academic (0)/Nonacademic (1)	2.27	1.12	0.86	0.29	0.32	0.44
Virus Uncertainty	13.57	7.68	5.58	2.63	3.47	2.39
Job Uncertainty	22.79	21.54	14.50	8.39	8.43	6.63
Workload Uncertainty	11.78	13.74	4.24	3.43	6.24	6.22
Logistics Uncertainty	11.18	10.71	4.63	4.45	4.22	1.84
Support Uncertainty	32.02	32.23	12.80	11.37	8.50	8.14
Past Well–Being			55.19	64.42	62.56	69.95
Past Well–Being × Virus Uncertainty			0.04	0.08	0.14	0.14
Past Well–Being × Job Uncertainty			0.12	0.48	0.38	0.15
Past Well–Being × Workload Uncertainty			0.47	0.70	0.54	0.20
Past Well–Being × Logistics Uncertainty			0.11	0.22	0.72	0.56
Past Well–Being × Support Uncertainty			0.26	1.59	0.21	0.39

Values have been rescaled to sum up to 100.

## Data Availability

Data are available from the corresponding author.

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
