# Peer review of "Uncertainty and Well-Being amongst Homeworkers in the COVID-19 Pandemic: A Longitudinal Study of University Staff"

_ijerph, 2022, doi:10.3390/ijerph191610435_

Round 1

Reviewer 1 Report

Review ijerph-1821844

Overall the paper is great. I like the approach, and the research design and methods used are good. There is some need for further explain some of the analyses and what statistics was used in the analyses - please see my comments below.

I am very happy with the measurement invariance testing that is done - only minor changes to the presented results would improve the dissemination of the results. 

I also have some comments on the interpretation of the results as the lack of concise findings over time, and the small effects indicate to me that the observed effects might be due to methodological and analytical artifacts (Common Method Variance, testing of multiple variable without prior expectations on specific effects of individual variables etc) rather than an actual relationship between the experienced Uncertainty and the Well-being of the participants. Please see my comments. 

Here are my comments: 

Abstract: Good abstract, but some long sentences  with multiple commas that can be shortened in order to ease readability. I leave to the authors to consider this point. 

Introduction: Complete and well written introduction. 

Research Question/Hypotheses: Well written, and well argued for. I wonder whether there should be some consideration of family-wise error rate and/or overfitting as multiple scales were used in multivariable regression analyses with two outcomes. This increases the probability that some of the scales will have a significant association with the dependent variable by chance - even when we use a multivariable linear regression analysis. The fact that the same variables are not clearly significant on all time steps, but rather that different variables are significant at different time steps reinforces my understanding that the observed effects here could be just a result of common method variance and random variation. This is also related to the following problem:

Line 460-468: IN the discussion of common method variance - the authors say that CMV unlikely as not all types of uncertainty were associated with well-being. I do not see how this argument can hold. Common method variance will increase the observed correlations between variables, As seen from table 3:  All uncertainty variables have correlations with anxiety above the threshold of r > +/- 0.12 (mentioned in the footnote of the table), and most correlations are in the range of -.18 to -.30 (except for logistics uncertainty) however, none of these correlations are  above a moderate size, so the association is not particularly strong. 

Also, the presence of CMW does not necessitate that all the variables will be statistically significant in a multivariable linear regression which controls for the other variables in the analysis. I think it would be a safer bet to say that common method variance probably exist in the data due to the use of same method to collect data and then to acknowledge that we do not know the extent of this CMV. The authors then need to do analytical steps to control for different levels of CMV - to make it plausible that the observed small effects are not due to CMV. As can be seen in Table 5 almost none of the uncertainties remain significant after you include past measurements of the dependent variable, meaning that the uncertainties explain so little variance that it is not statistically observable by the statistical model with the current sample size.  

General Comment on methods #1 - might it be smart to make an overall measure of uncertainty, and then use that in the regression models instead of the five facets of uncertainty? I would guess that the global uncertainty measure would be more robust in regression analyses - despite the fact that the CFA showed that the model with five facets were better than the model with one facet. However, this do not hinder you from making a global score and then using this in regression analyses. 

Table 1: The last question have a very low loading on October 2020 (0.093) is this correct? (should it be 0.93 instead?). If this loading is correct, what does it say for the model fit?

Table 2: 1) for all model comparisons I suggest calculating Cohen´s w as an effect size indictor of the size of the change in the Chi-square. 2) I would also consider to calculate other fit indexes such as McDonald´s Noncentrality index or Gamma hat to assess the extent of change when going from configure, metric and scale invariance models. Suggestions for these other model fit indexes can  for example be found in Cheung, G. W., & Rensvold, R. B. (2002). Evaluating goodness-of-fit indexes for testing measurement invariance. Structural equation modeling, 9(2), 233-255.

Table 3: 1) Please replace the name «alpha» with the symbol for Cronbach´s alpha.  2) Also consider to have the table horizontally on a page to include the whole table on one page and so you don’t have to have the titles over two lines. 3) Two variable names occur twice - Anxiety-Contentment-July and Anxiety-Contentment - July (row 1, 2, 8 and 9)

Table 4, 1) in the model for Depression-Enthusiasm, Why is there an x after the name «Virus Uncertainty»? 2) Is the regression analysis run with the variables as observed variables (I.e. with variable means) or as latent variables? Please state how the analysis was done. If you have added the variables as latent contracts then please also show the measurement and path diagram for the analysis. 

I would like to see the box+line or path diagram of the statisical model(s) so I can visually see how the model was set up. The authors can preferably create an appendix which shows the path diagrams so in this way the layout and structure of the paper is not greatly changed. 

It could also be interesting to know how the regression analyses go if cross-lagged data is used, so predictors from time 1 against outcomes at time 2, instead of predictors at time 2 vs outcome at time 2 with control for outcome at time 1. 

I would also like to know whether the authors intend to share their data. This is not explained in the data sharing section of the paper. 

Author Response

Comments and Suggestions for Authors

This paper discusses the relationship between uncertainty and well-being amongst homeworkers in the Covid-19 pandemic. The topic is interesting. However, several issues should be addressed.

  • Several kinds of uncertainty are displayed in Table 1. However, the source of them should be pointed out. The table outlines the five types of uncertainty and the items used to measure them.

Table 1 outlines the items from our newly-developed measure. Each of the items is clearly listed under a particular type of uncertainty (e.g., ‘job uncertainty’, ‘workload uncertainty’). We are not sure in this context what is meant by the source of uncertainty – do you mean where do the specific items come from (we developed the items ourselves, which we state in the text)? We are happy to elaborate, subject to guidance about what specifically you would like further information on. 

 (2)   In this paper, the authors give a detailed description on uncertainty amongst homeworkers in the Covid-19 pandemic. The analysis methods (or the quantitative analysis formula) used in this paper should be given.

      We included a detailed section on the analytic methods adopted in this research in our method section (entitled ‘Analysis approach’). We have added a further reference to these in the introduction of the manuscript to provide clarity from early in the paper:

“We then report a study that first develops measures of the five dimensions of uncertainty and then assesses through factor analysis if data on these confirms they are discrete, before testing, using structural equation modeling. the relationship between the types of uncertainty and well-being, and finally, using relative weights analysis we answer the open-ended question of which, if any uncertainty, may have the strongest association.”

If further information on the analysis is required, we would be happy to elaborate.

(3)   To strengthen the motivation of the paper, the characteristics of the study should be listed in Introduction section. Also, comparative analysis is necessary to demonstrate the advantages of the presented methods.

We have added the following details but if more specific characteristics are needed, please suggest them. The study is longitudinal using data from three time periods during the Covid-19 pandemic, two in 2020 and one in early 2021, collected from academic and professional services employees in two universities in England, with an initial sample of 584, and subsequent ones being of 394, and 384.

In including the Relative Weights Analysis we are offering a comparative analysis.

(4)   The Discussion should be divided into two sections, and the managerial implications can be offered.

We have now included section headings and added a section on practical implications, outlining the implications of our study for managers and policy makers.

(5)   Uncertainty exists here and there in the real world. For example, fuzzy uncertainty should be considered and handled in the Covid-19 pandemic. The following reference is useful for future research: Novel correlation coefficient between hesitant fuzzy sets with application to medical diagnosis. Expert Systems with Applications, 2021, 183: 115393.

Thank you for this interesting suggestion. Our focus in the present research is on subjective uncertainty, that is, individuals’ perceptions of externalities as being uncertain. Our assertion is that people’s subjective experiences reflect their appraisals of external situations and these in turn shape well-being.    

Reviewer 2 Report

This paper discusses the relationship between uncertainty and well-being amongst homeworkers in the Covid-19 pandemic. The topic is interesting. However, several issues should be addressed.

(1)   Several kinds of uncertainty are displayed in Table 1. However, the source of them should be pointed out.

(2)   In this paper, the authors give a detailed description on uncertainty amongst homeworkers in the Covid-19 pandemic. The analysis methods (or the quantitative analysis formula) used in this paper should be given.

(3)   To strengthen the motivation of the paper, the characteristics of the study should be listed in Introduction section. Also, comparative analysis is necessary to demonstrate the advantages of the presented methods.

(4)   The Discussion should be divided into two sections, and the managerial implications can be offered.

(5)   Uncertainty exists here and there in the real world. For example, fuzzy uncertainty should be considered and handled in the Covid-19 pandemic. The following reference is useful for future research: Novel correlation coefficient between hesitant fuzzy sets with application to medical diagnosis. Expert Systems with Applications, 2021, 183: 115393.

Author Response

Overall the paper is great. I like the approach, and the research design and methods used are good. There is some need for further explain some of the analyses and what statistics was used in the analyses - please see my comments below.

Thank you for your positive and encouraging verdict.

I am very happy with the measurement invariance testing that is done - only minor changes to the presented results would improve the dissemination of the results. 

I also have some comments on the interpretation of the results as the lack of concise findings over time, and the small effects indicate to me that the observed effects might be due to methodological and analytical artifacts (Common Method Variance, testing of multiple variable without prior expectations on specific effects of individual variables etc) rather than an actual relationship between the experienced Uncertainty and the Well-being of the participants. Please see my comments. 

Here are my comments: 

Abstract: Good abstract, but some long sentences  with multiple commas that can be shortened in order to ease readability. I leave to the authors to consider this point. 

We have broken the one long sentence up into two thus:

“Our empirical tests, show that uncertainties around the virus, employer support and their job quality have the strongest negative associations with well-being. These are based on data collected over three time periods in the first year of the pandemic from a sample of university staff (academics and non-academics) and well-being is measured on two continua, anxiety–contentment and depression–enthusiasm.”

Introduction: Complete and well written introduction.

Thanks for this judgement 

Research Question/Hypotheses: Well written, and well argued for. I wonder whether there should be some consideration of family-wise error rate and/or overfitting as multiple scales were used in multivariable regression analyses with two outcomes. This increases the probability that some of the scales will have a significant association with the dependent variable by chance - even when we use a multivariable linear regression analysis. The fact that the same variables are not clearly significant on all time steps, but rather that different variables are significant at different time steps reinforces my understanding that the observed effects here could be just a result of common method variance and random variation. This is also related to the following problem:

Line 460-468: IN the discussion of common method variance - the authors say that CMV unlikely as not all types of uncertainty were associated with well-being. I do not see how this argument can hold. Common method variance will increase the observed correlations between variables, As seen from table 3:  All uncertainty variables have correlations with anxiety above the threshold of r > +/- 0.12 (mentioned in the footnote of the table), and most correlations are in the range of -.18 to -.30 (except for logistics uncertainty) however, none of these correlations are  above a moderate size, so the association is not particularly strong. 

Also, the presence of CMW does not necessitate that all the variables will be statistically significant in a multivariable linear regression which controls for the other variables in the analysis. I think it would be a safer bet to say that common method variance probably exist in the data due to the use of same method to collect data and then to acknowledge that we do not know the extent of this CMV. The authors then need to do analytical steps to control for different levels of CMV - to make it plausible that the observed small effects are not due to CMV. As can be seen in Table 5 almost none of the uncertainties remain significant after you include past measurements of the dependent variable, meaning that the uncertainties explain so little variance that it is not statistically observable by the statistical model with the current sample size.  

Concerning the first issue regarding overfitting and family-wise error rate, the analysis we used to test our hypotheses was structural equation modelling. We selected this analysis method as it is ideally suited to address questions where there are multiple predictors. Although we agree that there is still the potential issue of inflated family-wise error rate, we decided against making any corrections. To our understanding this is not a typical approach to SEM models and therefore guidance in this regard is scarce. Importantly, simple corrections such as the Bonferroni adjustment are founded on the assumption that the multiple tests are independent of each other. As the correlation table shows, this is not the case for our data and using such adjustments could have reduced type I error at the expense of type II.

Concerning the second issue of common method variance, we have made two key changes. First, we took analytic steps to estimate the extent of common method bias in our data using the Harman single factor approach. We report these results at the end of section 6.1. We also tried the assess common method variance using the unmeasured common latent factor approach but unfortunately our results were inconclusive as we run into convergence issues due to a non-positive matrix.

Second, we have reworded our treatment of this in the discussion to explain more clearly that common methods variance is less likely to have been a problem as (a) some of our data and associations were from different timepoint, (b) we have found interaction effects that are more difficult to detect if common method variance were present, as well as to reflect our findings regarding the Harman single factor test.

General Comment on methods #1 - might it be smart to make an overall measure of uncertainty, and then use that in the regression models instead of the five facets of uncertainty? I would guess that the global uncertainty measure would be more robust in regression analyses - despite the fact that the CFA showed that the model with five facets were better than the model with one facet. However, this do not hinder you from making a global score and then using this in regression analyses. 

One of the key contributions of our study, in our view, is that we not only estimate the importance of uncertainty in shaping well-being though the first year of the pandemic, but that we investigate which types of uncertainties are most influential. This is important for shaping policy and practice, as well as for enhancing our understanding of the links between uncertainties and well-being from an academic perspective. Our measurement and analytic approach are well-supported by our measurement model, as you have noted, which confirms that our new measure captures five distinctive forms of uncertainty. We would argue that the benefits of our measurement and analytic approach are substantial and that the paper is stronger if we maintain our current approach.

Table 1: The last question have a very low loading on October 2020 (0.093) is this correct? (should it be 0.93 instead?). If this loading is correct, what does it say for the model fit?

Thanks for pointing this out. We have now corrected the typo to 0.93.

Table 2: 1) for all model comparisons I suggest calculating Cohen´s w as an effect size indictor of the size of the change in the Chi-square. 2) I would also consider to calculate other fit indexes such as McDonald´s Noncentrality index or Gamma hat to assess the extent of change when going from configure, metric and scale invariance models. Suggestions for these other model fit indexes can  for example be found in Cheung, G. W., & Rensvold, R. B. (2002). Evaluating goodness-of-fit indexes for testing measurement invariance. Structural equation modeling, 9(2), 233-255.

Thank you for the suggestion. We have now included the Gamma-hat index for all the CFA models in Table 2. For the Cohen’s w we were unable to find any literature on how to estimate it from chi-square difference scores of multigroup SEM models. The approach for calculating this from chi-square values obtained from a contingency table does not seem to easily translate to those that we can obtain from the difference between two chi-square values that represent SEM model fit.

Table 3: 1) Please replace the name «alpha» with the symbol for Cronbach´s alpha.  2) Also consider to have the table horizontally on a page to include the whole table on one page and so you don’t have to have the titles over two lines. 3) Two variable names occur twice - Anxiety-Contentment-July and Anxiety-Contentment - July (row 1, 2, 8 and 9)

  • Done
  • For us to do this would have mess up the journal template so we think it is better for the copy editor to do it.
  • Error corrected July changed to October.

Table 4, 1) in the model for Depression-Enthusiasm, Why is there an x after the name «Virus Uncertainty»? 2) Is the regression analysis run with the variables as observed variables (I.e. with variable means) or as latent variables? Please state how the analysis was done. If you have added the variables as latent contracts then please also show the measurement and path diagram for the analysis. I

  • x has been removed
  • We have used variable means rather than latent variables. We added a clarification in the second paragraph of section 5.

I would like to see the box+line or path diagram of the statistical model(s) so I can visually see how the model was set up. The authors can preferably create an appendix which shows the path diagrams so in this way the layout and structure of the paper is not greatly changed. 

While we appreciate that it can be helpful for readers to be able to visualise models, in the present case, the complexity of our models (particularly, although not exclusively, those including interaction effects) means that we think diagrams would not be helpful for readers in this instance. Given that we test multiple models (including with data at three time-points for the hypotheses), including diagrams for each model would also greatly expand the paper length. For this reason, we have chosen not to include diagrams.

It could also be interesting to know how the regression analyses go if cross-lagged data is used, so predictors from time 1 against outcomes at time 2, instead of predictors at time 2 vs outcome at time 2 with control for outcome at time 1. 

Our hypotheses explicitly predict immediate effects of uncertainty on well-being (hypothesis 1) and that uncertainty will predict a change in well-being (hypothesis 2). Our analytic approach of testing within-time-point effects (i.e., time 1 uncertainty predicts time 1 well-being) and change effects (i.e., time 2 uncertainty predicts time 2 well-being when time 1 well-being is controlled for) is therefore consistent with what we anticipate and predict, based on our theorising.

The suggestion here is to take a different approach to analysis, in which we would test if time 1 uncertainty predicts time 2 well-being. We agree that this can be an interesting and useful approach in general to test for effects over time with longitudinal data such as ours. However, we have reservations about such an approach for our particular study where the time periods between measures are long (July to October, October to February), given both the nature of our variables and the highly changing environment of the pandemic. For this reason, we would not be confident about the meaning of any associations evidenced with such an analytic approach and so have decided not to explore the data in this manner.

I would also like to know whether the authors intend to share their data. This is not explained in the data sharing section of the paper. 

We have completed this and all the other end points.

Reviewer 3 Report

This study reports on homeworkers and uncertainties during the COVID-19 pandemic, and the relationship between various types of uncertainties and wellbeing measures.  On of the strengths of this study is that it is longitudinal.  The researchers measure uncertainty and outcomes at three different times during the pandemic and assess uncertainty using items that measure uncertainty around the virus, job quality, workload, logistic  of worklives, and employer support.  They also examined the effects of prior wellbeing on current wellbeing.  The researchers conducted confirmatory factor analysis and decided upon a five factor solution to fit the items used to measure uncertainty along the different domains (job, workload, logistics, support, and virus uncertainty).  Findings indicated that uncertainty was positively associated with poor wellbeing.  Interestingly, uncertainty had a stronger negative effect for those reporting better prior wellbeing.  The paper is laid out well, and in an understandable manner.

Round 2

Reviewer 2 Report

The manuscript can be accepted. But the authors neglect the related reference in the following.   Uncertainty exists here and there in the real world. For example, fuzzy uncertainty should be considered and handled in the Covid-19 pandemic. The following reference is useful for future research: Novel correlation coefficient between hesitant fuzzy sets with application to medical diagnosis. Expert Systems with Applications, 2021, 183: 115393.

Author Response

Uncertainty is a subjective concept and not a property of the world external to the individual. We could not readily see the relevance of the article or the fuzzy concept to the research.